© Author(s) 2016. CC-BY 3.0 License.

- Relative role of individual variables on a revised Convective System Genesis 1
- Parameter over north Indian Ocean with respect to distinct background state.

<sup>1</sup>Sumesh. K.G, <sup>2</sup>Abhilash. S, <sup>3</sup>Ramesh Kumar. M. R.

- 1. Department of Agricultural Meteorology, University of Agricultural Sciences, Dharwad, 580005, India

2. Department of Atmospheric Sciences, Cochin University of Science and Technology, Cochin, 682016, India

3. National Institute of Oceanography, Dona Paula, 403004, India

Correspondence to: Sumesh.K.G. (sumeshgovind@gmail.com)

Abstract. Tropical storms are intense low pressure systems that form over warm tropical ocean 13 basins. Depending upon the intensity, they are classified as depressions, cyclones and severe 14 cyclones. Northern Indian Ocean (NIO) is highly prone to intense tropical storms and roughly 5-7 15 tropical storms are forming over this basin every year. Various Cyclogenesis indices are used to 16 forecast these tropical storms over various basins including NIO. In this aspect we propose a 17 revised Convective System Genesis Parameter (CSGP) to identify regions favourable for storm 18 genesis. The revised CSGP is constructed by using different combinations and thresholds of five 19 variables namely, the Low Level Relative Vorticity, the Low Level Convergence, the Shear coefficient, the Convective Instability parameter and the Humidity parameter. The relative role of 20 21 each individual variable on CSGP is analysed separately for different categories of the storms over 22 both Arabian sea and Bay of Bengal. The composite structure of the CSGP for different categories 23 of the storms is further evaluated separately for distinct large scale background state. The results 24 show that the revised CSGP is capable of distinguishing different categories of the storms. The 25 CSGP exhibits large variability during distinct large scale background state. It is also found that the 26 individual variables contribute in a different way during monsoon and non-monsoon seasons. The 27 revised CSGP can be used to forecast all categories of convective systems such as depressions, 28 cyclones and severe cyclones over NIO during the monsoon as well as non-monsoon seasons.

29

#### 30 1 Introduction

© Author(s) 2016. CC-B

31

Tropical storms are intense low pressure systems that form over warm tropical ocean basins. India 32 33 Meteorological Department (IMD) has classified these convective systems that form over NIO 34 depending upon their intensity based on the wind speed criteria. The systems are broadly classified 35 as depressions, cyclones and severe cyclones. Table 1. shows the classification of convective systems by IMD by using the wind speed criteria. If the wind speed of the convective system is 36 between 32-50 kmph it is called a depression and the wind speed is between 50-59 kmph it is called 37 38 a deep depression. These depressions and deep depressions produce good amount of rainfall in the 39 coastal and inland areas of the Indian sub continent. The frequency of these depressions and deep 40 depressions are more in the Bay of Bengal (BB) compared to the Arabian Sea (AS). Intense 41 systems are further classified into matured and developing convective systems such as cyclones 42 (wind speed between 60-90 kmph), severe cyclones (90-119 kmph), very severe cyclones (119-220 43 kmph) and super cyclones (> 220 kmph). Generally all the categories of the convective systems 44 except the depressions and deep depressions are called the tropical cyclones in various tropical 45 ocean basins. These tropical cyclones cause severe damage to the structures near the coastal areas 46 due to high wind and storm surge during their land fall. Though the origin of the tropical cyclones are not fully understood, studies have shown that there are few environmental parameters such as 47 48 Sea Surface Temperature (SST), Low Level Relative Vorticity (LLRV), Low Level Convergence 49 (LLC), Vertical Wind Shear (VWS), Middle Tropospheric Relative Humidity (MTRH), Convective 50 Instability (CI) and Middle Tropospheric Instability (MTI) known to control the formation and further intensification of a tropical cyclone (Gray., 1968; 1998; Palmen., 1948; Gray et.al., 1975). 51

52

Various authors have discussed that sea surface temperature and ocean thermal energy play 54 an important role in the formation and existence of the tropical cyclones. And it is observed that tropical cyclones form over the warm oceanic regions where the sea surface temperature is higher 55 56 than 26°C Palmen (1948). The importance of thermal buoyancy from the surface to middle levels 57 for cumulonimbus convection has been discussed by (Palmen., 1948, 1957). Tropical cyclones 58 generally do not form near the equator, because it requires a minimum magnitude of Coriolis force 59 for its genesis. Studies and observations show that the frequency of cyclone genesis is more over 60 the regions where seasonal value of middle-level humidity is high. The process of initiation of 61 sustained low level circulation centre is called cyclogenesis. Gray., (1968) has discussed about the relative roles of various air-sea interaction parameters for the initiation and intensification of the 62 63 tropical cyclones and found that, strong low-level relative vorticity and small vertical shear of the 64 horizontal wind play an important role in the formation and intensification of tropical cyclones.

### 65

Gray., (1975) introduced six primary genesis parameters for the formation of tropical 66 67 cyclones. This study stated that the seasonal tropical cyclone frequency is directly linked to a 68 combination of these six physical parameters and is a function of seasonality on a climatological 69 basis. Which thereafter referred to as primary climatological genesis parameters. These parameters are (1) Low level relative vorticity, (2) Coriolis parameter, (3) The inverse of the vertical shear of 70 71 the horizontal wind between the lower and upper tropsphere, (4) Ocean thermal energy or sea 72 temperature above  $26^{\circ}$ C to a depth of 60m, (5) Vertical gradient of equivalent potential temperature 73 between the surface and 500mb level and (6) Middle troposphere relative humidity. The first three 74 parameters are called the dynamic potential and the last three parameters are called the thermal 75 potential. The product of the dynamic and thermal potentials is referred to as the Seasonal Genesis 76 Parameter (SGP).

The SGP is usually calculated from the seasonally averaged climatological atmospheric or 78 79 oceanic fields for each of the four three month seasons: winter (JFM), spring (AMJ), summer (JAS) 80 and autumn (OND). Gray., (1975) has also proposed a Yearly Genesis parameter (YGP) and it is 81 calculated as the sum of the four SGPs in four seasons. The thermal potential of the SGP delimits 82 the regions and the seasons of possible tropical cyclone formation. The dynamical factors 83 determine the synoptic conditions favourable to the formation of tropical cyclones. The YGP which 84 incorporates both thermal and dynamical parameters is able to identify the regions having a high 85 probability of tropical cyclone formation on climatological time scales. Gray., (1979) validated 86 YGP against observations of the reported detection locations of storm systems which later became 87 tropical cyclones, according to WMO criteria, during 1958-1977. The calculations made by Gray are based on climatological observations averaged over the same period, and have shown that the 88 89 SGP is able to reproduce seasonal frequency distribution of observed tropical cyclones and their 90 geographical distribution over the different ocean basins. In the northern hemisphere the average 91 cyclogenesis is reasonable (but slightly over estimated in the northwest Pacific in spring and 92 Autumn). In the southern hemisphere, cyclone frequency is over estimated by the YGP especially 93 in southern Indian Ocean and south west Pacific. Royer et.al. (1998) modified Gray's YGP by 94 replacing the thermal potential with the convective potential. The convective potential is defined 95 as; "Convective Potential = k x  $P_c$ " over the oceans and for  $|\emptyset| = 35^\circ$  lat. Where  $P_c$  is the seasonal 96 mean convective precipitation in mm/ day computed by the model, and Ø is the latitude, and the 97 numerical value of k is 0.145. This modified YGP is called the Convective Yearly Genesis 98 Parameter (CYGP), which is found as successful as the original YGP for reproducing the main

99 areas of tropical cyclone genesis.

### 100

McBride and Zehr, (1981) introduced a Daily Genesis Parameter (DGP). This parameter is 102 calculated as the difference of relative vorticity at the upper level (200mb) and the lover level 103 (900mb). It is defined as (900 mb - 200 mb). This parameter could describe that (1) both non-104 developing and developing systems are warm core in the upper levels. The temperature (and 105 height) gradients are more pronounced in the developing systems, but the magnitudes are so small 106 that the differences would be difficult to measure for individual systems. (2) the developing or pre-107 typhoon cloud cluster exists in a warmer atmosphere over a large horizontal scale. (3) there is no 108 obvious difference in vertical stability for moist convection between the systems. (4) there is no obvious difference in moisture content or moisture gradient (5) pre-typhoon and pre-hurricane 109 110 systems are located in large areas of high values of low level relative vorticity. The low level 111 vorticity in the vicinity of a developing cloud cluster is approximately twice as large as that 112 observed with non developing cloud clusters. (6) Mean divergence and vertical motion for the 113 typical western Atlantic weather system is well below the magnitudes found in pre-tropical storm 114 systems. (7) Once a system has sufficient divergence to maintain 100 mb or more per day upward 115 vertical motion over a  $4^{\circ}$  radius area, there appears to be no relationship between the amount of 116 upward vertical velocity and the potential of the system for development. (8) cyclogenesis takes 117 place under conditions of zero vertical wind shear near the system center. (9) There is a 118 requirement for large positive zonal shear to the north and negative zonal shear close to the south of 119 a developing system. There is also a requirement for southerly shear to the west and northerly shear 120 the east. The scale of this shear pattern is over a  $10^{\circ}$  latitude radius circle with maximum amplitude 121 at  $\sim 6^{\circ}$  radius.

Zehr., (1992) introduced a parameter called Genesis Parameter (GP) to quantify the 124 cyclogenesis over the north-west Pacific Ocean. GP is the product of three dynamic parameters 125 such as Low Level Relative Vorticity at 850 hPa, Low Level Convergence (negative of Divergence) 126 at 850 hPa and Shear Co-efficient. This study showed that this genesis parameter was useful in 127 differentiating between the non-developing and developing systems in the western North Pacific. 128 Roy Bhowmic (2003) used this Genesis Parameter to study the developing and non-developing systems over NIO, and observed GP values around (20.0)10<sup>-12</sup> S<sup>-2</sup> against T-No: 1.5 has the 129 potential to develop into a severe cyclonic storm. Kotal et.al. (2009) proposed a cyclone genesis 130 131 parameter for the Indian Seas, termed as the Genesis Potential Parameter (GPP). This parameter is

defined as the product of four variables, namely vorticity at 850 hPa, middle tropospheric relative humidity, middle tropospheric instability and inverse of vertical wind shear. The parameter is 133 134 tested with a sample dataset of 35 non-developing and developing low pressure systems that formed 135 over the Indian Seas during the period of 1995-2005. The result shows that there is a distinction 136 between GPP values is found to be around three to five times greater for developing systems than 137 for non-developing systems. The analysis of the parameter at early development stage of a cyclonic 138 storm appears to provide a useful predictive signal for intensification of the system.

Philander, (1985) discussed about the El-Niño Southern Oscillations (ENSO) phenomena 141 occurring over the tropical Pacific Ocean basin which affects the global climate through 142 teleconnection. This is mainly influenced by the weakening and strengthening of trade winds over 143 the tropical belt through the modulation of Walker circulations. The amplitude of fluctuations in 144 the trade winds over the Pacific Ocean is associated with the abnormal warming or cooling of the 145 sea surface over eastern Pacific Ocean. The trade winds and Walker circulation in turn gets 146 modulated through the abnormal warming (cooling) over eastern Pacific Ocean known as El-Niño 147 (La-Niña). Scientists have observed that varius ocean atmospheric processes over the globe are 148 affected by these El-Niño and La-Niña events. The frequency variations of tropical cyclones over 149 various basins during the El-Niño and La-Niña years have been studied by Nakazawa., (2001), 150 Cahn., (1985), Chia and Ropelewski, (2002) and Dong., (1988) The results show that the 151 frequency of tropical cyclones is found to be more during the El-Niño years over some basins and 152 in some other basins the frequency is more during the La-Niña years.

Ashok etal., (2007) proposed the new type of El-Niño and La-Niña events which is different 155 from the canonical El-Niño and La-Niña conditions. These events are termed as El-Niño Modoki 156 and La-Niña Modoki. The El-Niño Modoki event is defined as the warmer sst's in the central 157 pacific ocean and cooler sst's in both east and west pacific ocean. And a La-Niña Modoki event is 158 defined as the cooler sst's in the central pacific ocean and warmer sst's in the east and west pacific 159 ocean. Sumesh and Ramesh Kumar., (2013) have studied the influence of El-Niño Modoki events 160 on the tropical cyclones over north Indian Ocean. They observed that there are more cyclones over 161 AS during the El-Niño Modoki years than the El-Niño years. And in the case of severe cyclones 162 over AS the frequency is more during the El-Niño Modoki years than the El-Niño years, where as in 163 BB the frequency is more in the El-Niño years than the El-Niño modoki years.

Natural Hazards and Earth System Sciences Discussions

The present study is an attempt to evaluate the relative contribution of potential dynamical 166 and thermo dynamical parameters in the formulation of the revised CSGP. The main objectives of 167 the study are to quantify the composite variation of the revised CGSP over NIO. The cyclogenesis 168 over a particular basin is mostly linked to warm SST boundary and associated large scale 169 circulation pattern. Hence, the variation of this CGSP and individual parameters are studied with 170 respect to distinct background state. In this present study we have studied the frequency variations 171 of all the convective systems over NIO during the El-Niño, El-Niño Modoki, La-Niña, La-Niña 172 Modoki, Positive IOD, Negative IOD years. We have also selected some years as Neutral years in 173 which there were no significant warming or cooling of ocean water over equatorial Pacific or Indian 174 Ocean basin. 175 176 2 Methods and data used 177 2.1 The revised Convective System Genesis Parameter (CSGP) 178 179 180 The Convective System Genesis Parameter (CSGP) is a new modified index and it is different from 181 the Genesis Parameter (GP) defined by Zehr (1992) and the Genesis Potential Parameter (GPP) 182 defined by Kotal (2009). We use the dynamical parameters defined by Zehr (1992), the humidity 183 parameter defined by Kotal (2009) and the Convective Instability parameter defined by Gray, 184 (1975). Hence the revised index is a product of five parameters and it is defined as 185 186  $CSGP = (850VOR \cdot 850LLC \cdot S \cdot HUM \cdot CI)$ 187 188 Where 189 190 1) 850VOR= Low Level Relative Vorticity at 850 hPa (LLRV) 191 2) 850LLC= Low Level Convergence (negative of Divergence) at 850 hPa (LLC) 192 3) VWSC= Shear Co-efficient = 25.0ms-1 - (200-800 SHEAR) / 20.0ms-1 (SHR)

- 4) HUM = [RH 40] / 30, Middle tropospheric relative humidity. (Where RH is the mean relative
- humidity between 700 and 500 hPa)
- 5) CI = (ThetaE\_1000 ThetaE\_500), Vertical gradient of Equivalent potential temperature,
- between 1000hPa and 500hPa.

Here in the Shear parameter we have kept the maximum magnitude as 25ms<sup>-1</sup>, the magnitude 198 greater  $25 \text{ms}^{-1}$  it will reduce CSGP to zero. The unit of this index is  $10^{-10} \text{s}^{-2}$  degree K. In this study 199 200 we have analysed all the convective systems that formed over NIO during the study period (1979-201 2008). We have also classified the convective systems in three categories as (1) depressions (which 202 include both depressions and deep depressions), (2) cyclones (which include only cyclones) and (3) 203 severe cyclones (which include severe cyclones, very severe cyclones and super cyclones). And we 204 have analysed the characteristics of this Index (CSGP) for all the categories in both monsoon 205 (JJAS) and non monsoon (JFMAM-OND) months.

2.2 Datasets used and selection of distinct background state

We have used NCEP/NCAR- Reanalysis 2 (Kanamitsu et.al., 2002), atmospheric data set (daily 210 mean) to calculate and analyse the dynamic as well as thermodynamic parameters. These 211 cyclogenesis parameters are averaged over the period of the convective systems. The spatial 212 resolution of this data is 2.5 x 2.5-degree grid. We have considered the whole NIO basin covering 213 the area bounded by  $50^{\circ}$ E to  $100^{\circ}$ E, and  $0^{\circ}$  to  $25^{\circ}$ N. In the present study, composite of all the 214 individual parameters during the period of the each convective systems over NIO for all distinct 215 basic states namely the El-Niño, El-Niño Modoki, La-Niña, La-Niña Modoki and both negative and 216 positive phases of Indian Ocean Dipole (IOD) and the Neutral years have been evaluated. We have 217 further divided the convective systems as they have formed in the monsoon and non-monsoon 218 seasons. The individual parameters and CSGP in the monsoon and non-monsoon seasons is then 219 composited separately irrespective of the large scale background state. Spatial correlations have 220 been computed between each of the individual parameters with the CSGP. Further we have 221 computed the correlation between each parameter and CSGP averaged around the convective 222 system.

#### 224 3 **Results and discussion**

3.1 Grouping the storms into different categories of basic state

In the present study we have analysed the characteristics of the Convective System Genesis Parameter (CSGP), over NIO for all the cases of convective systems such as (1) Depressions, (2) 229 230 Cyclones and (3) Severe cyclones. The entire study period is divided into El-Niño, El-Niño 231 Modoki, La-Niña, La-Niña Modoki, Positive IOD, Negative IOD and the Neutral years. We have 232 again sub divided the convective systems for their formation in the monsoon seasons as well as the 233 non-monsoon seasons of each years. The ENSO and IOD years we have take from the recent 234 publications. To avoid the combined effects of ENSO and IOD and we have selected these years in 235 such a way that the ENSO and IOD activities are not occurring simultaneously. And we have also 236 selected some years as the Neutral years, in which there were no establishment of either ENSO 237 activity over the Pacific Ocean or an IOD activity over the NIO. Table 2. Shows the selected 238 ENSO and IOD activity years during the study period. There were four El-Niño years, seven El-239 Niño Modoki yeras, six La-Niña years, two La-Niña Modoki years, one Positive IOD year and three 240 Negative IOD years during the study period. Table 3. Shows the total frequencies of the 241 convective systems formed over NIO during the study period. In the case of depressions over AS, 242 there have been 10 depressions during the monsoon seasons and 17 depressions during the non-243 monsoon seasons, and in the case of depressions over BB, there have been 63 depressions during 244 the monsoon seasons and 44 depressions during the non-monsoon seasons of the study period. In 245 the case of cyclones over AS, there have been 3 cyclones during the monsoon seasons and 5 246 cyclones during the non-monsoon seasons, and in the case of cyclones over BB, there have been 8 247 cyclones during the monsoon seasons and 27 cyclones during the non-monsoon seasons of the 248 study period. And in the case of severe cyclones over AS, there have been 5 severe cyclones during 249 the monsoon seasons and 11 severe cyclones during the non-monsoon seasons, and in the case of 250 severe cyclones over BB, there have been 3 severe cyclones during the monsoon seasons and 42 251 severe cyclones during the non-monsoon seasons of the study period

252

The black and grey histograms in figure 1. Shows total number and per year count (frequencies) of 254 the depressions over NIO during the study period. From figure 1(a), the black histograms show the 255 frequencies and per year values of the depressions over AS. It is observed that, the frequency is

Natural Hazards and Earth System Sciences

more during the El-Niño, El-Niño Modoki, La-Niña and the Neutral years, and it is less during the 257 PIOD and NIOD years. And the gray histograms, show the per year values of the depressions over 258 AS during the study period. It is observed that the per year value is more (greater than 1 depression 259 per year) for the El-Niño, La-Niña, PIOD and Neutral years, and the per year value is less for the 260 El-Niño Modoki and NIOD years. From figure 1 (b), the black histograms show the frequencies of 261 the depressions over BB. It is observed that, the frequency is more during the El-Niño, El-Niño 262 Modoki, La-Niña, NIOD and Neutral years, and it is less during the La-Niña Modoki and PIOD 263 years. The Irrespective of the background state, all per year value is found to be greater than 3.0 for 264 the depressions over BB. However, both IOD years shows more favourable condition for the 265 depression to form over BB.

Further we analysed the depression frequencies separately for monsoon and non-monsoon seasons. 268 Figure 2, gives the seasonal frequencies of depressions and its count per season over NIO during the 269 monsoon and non-monsoon seasons. From figure 2 (a), the black histograms show the number of 270 depressions over AS during the monsoon seasons. It is observed that, the La-Niña years show 271 maximum frequency with per season values close to 1(0.83). And the minimum frequency is 272 observed during the El-Niño Modoki, NIOD and Neutral years. It is evident from the figure that all 273 other background states are not favourable for the formation of depression over AS during monsoon 274 months. The black histograms in figure 2 (b) shows the number of depressions over AS during the 275 non-monsoon seasons. The maximum number is observed during the El-Niño Modoki years and the 276 minimum frequency is observed during the PIOD years. And it is also noticed that a good number 277 of depressions have formed during the El-Niño, La-Niña and the Neutral years. The gray 278 histograms show the per seasonal values for the depressions oer AS. It is observed that, the per 279 seasonal values are more for the El-Niño, PIOD and Neutral years, and it is less for the El-Niño 280 Modoki and La-Niña years. From figure 2 (c) shows the number of depressions over BB during 281 the monsoon seasons. It is observed that, the depression count is more during the El-Niño, El-Niño 282 Modoki, La-Niña and the Neutral years. And the frequency is less during the La-Niña Modoki, 283 PIOD and NIOD years. It is clear from the figure that the per seasonal values of depression over 284 BB during monsoon season is more (greater than 3 per season) during El-Niño and PIOD years. The 285 per seasonal count is close to 2 for all other categories of years during monsoon season. From figure 286 2 (d), the black histograms show the number of depressions over BB during the non-monsoon 287 seasons. It is observed that, the El-Niño Modoki years are having the maximum frequency, and the 288 La-Niña Modoki and PIOD years are having minimum frequency. And it is also observed that a 289 good number of depressions have formed during La-Niña, NIOD and PIOD years. The gray

290 histograms show the per seasonal frequencies of depressions over BB during the non-monsoon 291 seasons. It is found that more than 2 depressions per season were formed during El-Niño Modoki, 292 PIOD and NIOD years. In general the frequencies of depressions over BB are more during the 293 monsoon seasons of El-Niño, El-Niño Modoki, La-Niña and Neutral years as compared to the non-294 monsoon seasons.

295

Figure 3, gives the total number and per year frequencies of the cyclones over NIO. From figure 3 297 (a), the black histograms show the count of cyclones over AS. It is observed that, the frequency is 298 more during the El-Niño Modoki and Neutral years, and it is less during the La-Niña and PIOD 299 years. The gray histograms show the per year values of the cyclones over AS. The per year values 300 of less than 1 shows that almost all categories of years are not conducive for the development of 301 cyclones over AS. From figure 3 (b), the black histograms show the total number of cyclones over 302 BB during different background states. It is observed that, more number of cyclones were formed 303 during the La-Niña and Neutral years, and it is less during the PIOD years. The gray histograms 304 show the per year values of the cyclones over BB. It is observed that, on an average more than 2 305 cyclones are formed during La-Niña Modoki and Neutral years.

Figure 4. Gives the seasonal and seasonal frequencies of cyclones over NIO during the monsoon 308 and non-monsoon seasons. It is seen from figure 4(a) that, only very few cyclones were formed over 309 AS during the monsoon seasons of El-Niño Modoki, PIOD and the Neutral years, and no cyclones 310 were formed over AS during the monsoon season of the other selected years. The gray histogram 311 shows frequency of the per seasonal values of the cyclones over AS during the monsoon seasons. 312 From figure 4 (b), shows the total number and seasonal frequencies of the cyclones over AS during 313 the non-monsoon seasons. It is observed that there were only very few (5) cyclones formed during 314 the non-monsoon months of El-Niño Modoki, Neutral and La-Niña years. From figure 4 (c) the 315 total number and seasonal frequency of cyclones over BB during monsoon moths are more during 316 La-Niña and NIOD years the seasonal frequency is less than 0.25 during El-Niño and Neutral years. 317 It is observed from figure 4(d) that, the frequency of occurrence of cyclones during non-monsoon 318 months are more during La-Niña Modoki and neutral years. And it is also observed that there were 319 good number of cyclones over BB during the non-monsoon seasons of El-Niño, El-Niño Modoki, 320 La-Niña, La-Niña Modoki years.

Figure 5. Gives total number and yearly frequencies of the severe cyclones over NIO. It is observed 323 that from figure (5a) the total number and the frequency of severe cyclones over AS is more during 324 the El-Niño Modoki, La-Niña, NIOD and Neutral years. On an average, PIOD and NIOD years 325 shows at least one severe cyclone per year. It is clear from figure 5(b) that the total number and the 326 frequency of occurrence of severe cyclone over BB is more during the La- Niña and El-Niño years 327 and it is less during the PIOD years. It is also observed that at least one severe cyclone have formed 328 over BB during all categories of background state. Figure 6. Shows the total number and seasonal 329 frequencies of the severe cyclones over NIO during the monsoon and non-monsoon seasons of the 330 study period. It is found from figure 6(a) that the total number and a seasonal frequency of severe 331 cyclones over AS during monsoon season are very low for all different categories of years. It is seen 332 from figure 6 (b) that the frequency of occurrence of Severe cyclone over AS during non-monsoon 333 month is maximum during El-Niño Modoki years., From figure 6 (c, it is clear that formation of 334 severe cyclone over BB is rare during monsoon months except for the two cyclones formed during 335 El-Niño and Neutral years. It is observed from figure 6(d) that total number and seasonal 336 frequencies of the formation of severe cyclone over BB during non-monsoon months are more 337 during La-Niña and El-Niño years. It is also noticed that a good number of severe cyclones have 338 formed during all selected years of different background state.

3.2 Spatial variation of CSGP with respect different seasons

Figures 7-9 presents variations of the CSGP for the convective systems over NIO during monsoon 343 as well as non-monsoon seasons of the study period. The categories of the convective systems have 344 been named as DD for depressions, CS for cyclones and VS for severe cyclones. And the genesis 345 locations are represented as black dots. Figure 7 shows the variations of CSGP for the depressions 346 formed over NIO. It is observed that the genesis points of the depressions over AS during the 347 monsoon seasons are clustered around the region of  $14^{\circ}N - 20^{\circ}N$  and  $64^{\circ}E - 72^{\circ}E$ . But during the 348 non-monsoon seasons the genesis points of the depressions are spread widely in the region of  $5^{\circ}N$  – 349  $20^{\circ}$ N and  $58^{\circ}$ E –  $77^{\circ}$ E. Whereas over BB, the genesis locations of the depressions during monsoon 350 season are clustered in the area of  $14^{\circ}N - 22^{\circ}N$  and  $83^{\circ}E - 93^{\circ}E$ . This region corresponds to the 351 eastern end of monsoon trough and large values of CSGP found along the monsoon trough region. 352 However, during the non-monsoon seasons the genesis locations spread over a large area of  $5^{\circ}N$  – 353  $20^{\circ}$ N and  $78^{\circ}$ E –  $97^{\circ}$ E. Most favourable genesis locations with higher values of CSGP is found 354 around the region of 5-15N and 83-90E. It is found that lower values of CSGP favours the