# Peer review of "© Author(s) 2016. CC-BY 3.0 License."

_Natural Hazards and Earth System Sciences, 2016_

## Referee Comment (RC1) · Anonymous Referee #1 · 19 Oct 2016

Review of the paper: Relative role of individual variables on a revised Convective System Genesis Parameter over north Indian Ocean with respect to distinct background state, by K. G. Sumesh, et al.

General comments: In this paper a revised Convective System genesis Parameter is presented together with an analysis of the different role of the parameters for different seasons. A lot of work has been done by the authors, but to completely appreciate their work, the readability of the paper has to be increased.

There are a few major weaknesses:

1) a too long introduction, there is no need to describe in such detail the work from

previous paper

2) there is no need for a detailed definition of well known meteorological events like ENSO etc.

3) there are several typos and English is poor.

I believe that after a careful and deep revision the paper can be accepted.

Specififc Comments:

page 4 lines-101-121: This part has to be summarized

page 5 lines 140- 152: as for point 1

page 10 line 300 shows change in show

page 10 line 315 moths do you mean months?

....and many others!